# Non-Invasive Quantification of Faecal and Urine Reproductive Hormone Metabolites in the Naked Mole-Rat (*Heterocephalus glaber*)

**DOI:** 10.3390/ani13193039

**Published:** 2023-09-27

**Authors:** Tshepiso Lesedi Majelantle, Andre Ganswindt, Stefanie Birgit Ganswindt, Nicole Hagenah, Daniel William Hart, Nigel Charles Bennett

**Affiliations:** Mammal Research Institute, Department of Zoology and Entomology, University of Pretoria, Hatfield, Pretoria 0028, South Africa; aganswindt@zoology.up.ac.za (A.G.); stefanie.ganswindt@up.ac.za (S.B.G.); nicole.hagenah@up.ac.za (N.H.); daniel.hart@zoology.up.ac.za (D.W.H.); ncbennett@zoology.up.ac.za (N.C.B.)

**Keywords:** GnRH administration, faecal androgen metabolites, faecal progestogen metabolites, urine androgen metabolites, urine progestogen metabolites, African mole-rats

## Abstract

**Simple Summary:**

The naked mole-rat occurs in colonies with a distinct dominance hierarchy which includes a dominant breeding female, from one to three breeding males, and a number of subordinate individuals that are physiologically blocked from reproducing by the breeders but can reproduce if removed from the colony. Due to their small size, blood sampling is limiting, and measuring reproductive hormones in faeces or urine is considered an alternative method of quantification. Thus, we aimed to validate enzyme immunoassays (EIAs) for measuring androgens and progestogens or their metabolites in naked mole-rat urine and faeces. We administered gonadotrophin-releasing hormone (GnRH), a molecule which stimulates the release of reproductive hormones, and a saline control to twelve (six males and six females) naked mole-rats. The results revealed that urine is possibly not ideal for measuring reproductive hormones in naked mole-rats as no signal administration was detected in the matrix. A 5α-Progesterone EIA and an Epiandrosterone EIA were identified as suitable for quantifying faecal progesterone and androgen metabolites in male and female naked mole-rats. In addition, differences in how the animals responded to a GnRH challenge suggest that some individuals of both sexes are still incapable of reproducing even after being separated from the colony and the breeding female.

**Abstract:**

The naked mole-rat (*Heterocephalus glaber*) occurs in colonies with a distinct dominance hierarchy, including one dominant, breeding female (the queen), 1–3 breeding males, and non-reproductive subordinates of both sexes that are reproductively suppressed while in the colony. To non-invasively evaluate reproductive capacity in the species, we first had to examine the suitability of enzyme immunoassays (EIAs) for determining progestogen and androgen metabolite concentrations in the naked mole-rat, using urine and faeces. A saline control and gonadotrophin-releasing hormone (GnRH) were administered to twelve (six males and six females) naked mole-rats which were previously identified as dispersers and housed singly. The results revealed that urine is possibly not an ideal matrix for progestogen and androgen metabolite quantification in naked mole-rats as no signal was detected in the matrix post GnRH administration. A 5α-Progesterone EIA and an Epiandrosterone EIA were identified as suitable for quantifying faecal progesterone metabolites (fPMs) and faecal androgen metabolites (fAMs) in males and females, respectively. The results suggest that there are individual variations in baseline fPM and fAM concentrations, and only two out of six females and no males exhibited an increase in fPM concentrations greater than 100% (−20% SD) post GnRH administration. Conversely, only four out of six females and three out of six males had an increase in fAM concentrations greater than 100% (−20% SD) following GnRH administration. These results imply that some naked mole-rat individuals have a reduced reproductive capacity even when they are separated from the queen.

## 1. Introduction

Sociality, the non-random formation of groups can include animal units with asymmetry in aggression (dominance) wherein the highest position has priority access to resources and reproduction [1]. In addition, some social systems include subordinate individuals that are able to reproduce but forfeit their own reproductive fitness [2]. As a consequence, this leads to a reproductive skew in a group in which most individuals are likely to spend their lives without reproducing [2,3,4]. For example, in cooperatively breeding pied babblers (*Turdoides bicolor*), a significant proportion of the group do not breed but provide alloparental care for young [2,3], while the dominant and monogamous breeding pair account for 95.2% of all chicks in their group [5]. A potential explanation for why reproductively capable subordinates which do not reproduce is inbreeding avoidance [6] or suppression by the dominant individuals [7]. The reproductive suppression of subordinates by dominant individuals arises as a consequence of social suppression such as aggressive interactions directed towards subordinates and/or physiological suppression [8]. For example, in cooperatively breeding African wild dogs (*Lycaon pictus*), there was no significant difference in sperm quality parameters between male dominants and subordinates; thus, the suppression is thought to be behavioural, with the dominant male blocking access to the dominant female [9]. The physiological reproductive suppression of subordinates by dominant individuals is likely linked to the hypothalamic–pituitary–gonadal axis [10].

When the hypothalamic–pituitary–gonadal axis is stimulated, the axis releases gonadotrophin-releasing hormone (GnRH) which, in turn, stimulates the release of follicle-stimulating hormones (FSHs) and luteinizing hormones (LHs), collectively referred to as gonadotrophins [11]. Thereafter, these gonadotrophins stimulate the release of oestrogens, progestogens, and androgens by the gonads into general circulation [12,13]. After the hormones are delivered to their respective tissues, circulating hormones are subsequently metabolised by the liver and excreted, often as conjugates, via the kidneys into the urine or via bile into the gut [14].Thus, the related oestrogen, progestogen, and androgen metabolite concentrations can be measured non-invasively in urine and faeces. The non-invasive measurement of these metabolites is useful in small mammals since blood sampling is limited in quantity and frequency [15]. However, non-invasive steroid quantification for a species studied for the first time must be reliably validated [16].

African mole-rats (Bathyegidae) are subterranean mammals comprising six genera which range from solitary species (*Georychus*, *Bathyergus*, and *Heliophobius*) to truly social species (*Heterocephalus*, *Fukomys*, and *Cryptomys*) [17]. Within social African mole-rats, the naked mole-rat (*Heterocephalus glaber*) is considered eusocial, which is the most extreme sociality in mammals [18]. Species are considered eusocial if they have a reproductive division of labour, overlapping generations, and cooperative care for young [19,20]. Naked mole-rats live in colonies of up to 80 individuals with only one breeding female and from one to three breeding males, and the rest are non-reproductive subordinates [18]. These non-reproductive individuals are identified as helpers who perform tasks such as nest building, burrowing, and food carrying [18,21]. Furthermore, non-reproductive subordinates include disperser morphs which persistently attempt to disperse and prefer pairing up with unrelated individuals or joining unrelated colonies [22,23].

Non-reproductive female subordinates are anovulatory and show little or no follicular development [24], while impaired spermatogenesis in males is possibly due to reduced GnRH secretion [25,26]. However, within a short time after removal from their natal colony, their progestogen (in females) and androgen (in males) concentrations (measured in plasma and urine) become similar to those of breeders, and the individuals become reproductively viable [25,26,27]. In this study, we aimed to validate the suitability of enzyme immunoassays (EIAs) for determining progestogen and androgen metabolite concentrations in the naked mole-rat, using urine and faeces as a hormone matrix. For this, a saline control and exogenous GnRH (2 µg) were administered to twelve (six males and six females) naked mole-rats which had been previously identified as dispersers based on their escape attempts from the natal colony and were subsequently housed singly.

## 2. Materials and Methods

### 2.1. Study Animals and Housing

This study used twelve (six males and six females) captive-bred, individually housed naked mole-rats. The individuals had a mean body mass of 42.88 ± 2.18 g and were housed individually in plastic chambers (length = 35 cm, width = 30 cm, and height = 20 cm). Nesting material comprising sterilised wood shavings that were provided, in addition to a small opaque, plastic box with one side opening to serve as a nest (length = 15 cm, width = 10 cm, and height = 5 cm). The separated animals could not communicate visually or through olfaction with neighbouring individuals. These animals had been separated from their natal colony for a period lasting between 119 and 897 days based on their body mass and frequency of escape attempts from the natal colony [22]. The naked mole-rats were kept in a room maintained on a 12L:12D light schedule at 29–30 °C and a humidity of 40–60% at the University of Pretoria. The animals were fed sweet potatoes, cucumbers, apples, and bell peppers ad libitum. As naked mole-rats derive all water from their food resource, no additional free water was provided [28]. The study was approved by the University of Pretoria Ethics Committee (Project number NAS199/2020).

### 2.2. Sampling Schedule and Administration

Individual faecal and urine samples were collected over an experimental period of 21 days (5 days prior to and 5 days after saline and GnRH administration respectively, with one day of rest between experiments). On day 5 of the experiment, all 12 animals received a saline (control) injection (200 µL of sterile isotonic saline) subcutaneously. On day 16 of the experiment, all the animals were administered GnRH (2.0 µg of exogenous GnRH in 200 µL of sterile isotonic saline) subcutaneously.

Faeces and urine were sampled between 12 November and 4 December 2020. At the beginning of each sampling day (08:00–15:00), each animal was temporarily removed from its enclosure (maximum 5 min) to clean the housing. For this, the wood shavings were removed, and the enclosure was wiped with 70% ethanol before the animal subsequently returned to the enclosure. Thereafter the enclosures were checked every 30 min, and all freshly voided faecal samples were collected using forceps (cleaned with 70% ethanol between each sample), whereas the urine samples were collected using individually labelled plastic pipettes which were cleaned between each sample by rinsing with distilled water and air-dried between each sampling event. Combined urine and faecal samples were discarded due to cross-contamination. At the end of the day, new wood shavings were placed in the enclosure. On administration days (17 November and 29 November at 10 a.m.), the individuals were checked every 30 min for 30 h post administration, and all voided samples were collected. In total, 436 faecal samples (males = 222 samples; females = 214 samples) and 187 urine samples (males = 82 samples; females = 105 samples), were collected during the saline and GnRH administration period.

### 2.3. Sample Storage and Steroid Extraction

The urine samples were stored native at −20 °C until analysis. The faecal samples were lyophilised and pulverised, following the procedures described by Fraňková et al. [29]. Due to the small size of the individual faecal samples, samples with masses of 0.0150–0.0249 g, 0.0250–0.0366 g, and 0.0370–0.055 g were extracted using volumes of 0.5 mL, 1 mL, and 1.5 mL of 80% ethanol, respectively [30]. Consequently, samples less than 0.0150 g were considered too small [31] and were not used in any further analyses (*n* = 30). Thereafter, the suspensions were vortexed for 15 min and centrifuged at 1500× *g* for 10 min. The resulting supernatants were transferred into microcentrifuge tubes and stored at −20 °C until further analysis.

### 2.4. Enzyme Immunoassays

To validate the suitability of two progestogen and two androgen enzyme immunoassays (EIAs) for the non-invasive measurement of reproductive hormones, a subset of 111 faecal samples (3 males = 57 samples; 3 females = 54 samples) and 100 urine samples (6 males = 46 samples; 6 females = 54 samples) were analysed. Finally, the remaining faecal samples (325 samples) for both sexes (GnRH administration: 3 males = 65 samples and 3 females = 56 samples; saline administration: 6 males = 100 samples and 6 females = 104 samples) were measured for the reproductive hormones using the selected EIAs (see the Results section). Prior to the analysis, native urine and faecal steroid extracts were brought to room temperature and vortexed. After vortexing, the native urine samples were spun down for 15 s and a sample was taken just below the surface to avoid sediment at the bottom of the tube. Immunoreactive faecal progestogen metabolite (fPM) and urine progestogen metabolite (uPM) concentrations were determined using a (i) 5α-Progesterone EIA [32] and a (ii) Progesterone EIA [33]. Immunoreactive faecal androgen metabolite (fAM) and urine androgen metabolite (uAM) concentrations were determined using an (i) Epiandrosterone EIA [34] and (ii) a Testosterone EIA [34], according to the procedure described by Ganswindt et al. [35]. Details on the EIA antibodies, label, and standard are provided in the Appendix A. The sensitivities for the 5α-Progesterone EIA and Progesterone EIA were 320 pg/mL urine and 200 pg/mL urine, respectively, as well as 6.0 ng/g dry faecal weight (DW) and 9.6 ng/g DW, respectively. The sensitivities for the Epiandrosterone EIA and Testosterone EIA were 240 pg/mL urine and 40 pg/mL urine, respectively, and 7.2 ng/g DW and 1.2 ng/g DW, respectively. The coefficients of variation for intra-assay variance, determined by measuring high-quality (*n* = 18) and low-quality (*n* = 17) controls, created from respective diluted standards, were 5.55–5.73% and 4.36–6.67%, respectively, for the 5α-Progesterone EIA and Progesterone EIA, and 4.96–5.09%, and 5.57–6.67%, respectively, for the Epiandrosterone EIA and Testosterone EIA. The coefficients of variation for inter-assay variance, also determined by measuring high- and low-quality controls, measured three times in each and created from respective diluted standards, were 12.78–13.70%, 8.28–9.99%, 12.81–14.39%, and 8.72–14.78% for the 5α-Progesterone, Progesterone, Epiandrosterone, and Testosterone EIAs, respectively. All samples, standards, and quality controls were measured in duplicate. Serial dilutions of the faecal extracts resulted in displacement curves that were parallel to the respective standard curves and had relative variations in the slopes of the respective trend lines of <4% and <5% for the 5α-Progesterone EIA and Epiandrosterone EIA, respectively (Appendix A).

To achieve comparability for the steroid metabolite concentrations from the urine samples by expressing concentrations as mass per mg of creatinine, a modified Jaffe reaction was used to measure the creatinine concentrations in all native urine samples [36]. Native urine samples with creatinine concentrations of less than 0.05 mg/mL were considered diluted and consequently discarded (males = 19; females = 40). The steroid concentrations from the faecal samples were expressed per mass of dry faecal matter. All laboratory analyses were conducted at the Endocrine Research Laboratory, University of Pretoria, South Africa.

### 2.5. Data Analysis

All statistical analyses were carried out using the R program on the R studio interface RStudio (version 3.6.1) [37]. For each sex hormone, the most suitable EIA was identified based on the individual and collective increases in androgen and progestogen metabolite concentrations following the administration of GnRH. One male (M3) provided two urine samples during the GnRH administration period and six urine samples during the saline administration period and was thus considered to have a small sample size and removed from further analysis. For both matrixes, baselines for each individual naked mole-rat were calculated using all samples prior to administration and 48 h post administration for each sex hormone; these were used to calculate the percentage response to the administration of GnRH (Equation (1)). For each EIA, the peak sample within 48 h and the highest percentage change were selected.
(1)$$\frac{Sample\;concentration-Baseline}{Baseline}\times 100$$

Individual medians of the steroid metabolite concentrations for each time interval (12, 24, 48, 72, 96, and 120 h) for the GnRH and saline administrations were treated as categorical variables. For each sex, a linear mixed-effects model, taking the ID as a random effect, was used to analyse whether there were significant changes in the concentrations of fPM and fAM in relation to the interaction between the time interval and administration (Equation (2)), using the lme4 package [38].
*RH~Time: Administration + (1|ID)*(2)
where RH = the androgen or progestogen concentrations for each sex, time = categorical time intervals,: = an interaction, and administration = saline administration or GnRH administration.

The androgen metabolite concentrations for both matrixes and sexes and the progestogen metabolite concentrations for both matrixes in females had a positively skewed distribution and were thus log_10_ transformed prior to statistical analysis. For each model, normality was tested on the model residuals using quantile comparison plots and Levene’s test of homogeneity of variance. The results are presented as means ± standard errors (SEs), and differences between data sets were found to be significant at *p* < 0.05.

## 3. Results

### 3.1. EIA Validation

For urine samples, none of the six females and only one out of the five males showed an increase greater than 80% post GnRH administration in the uAM concentrations for both the Epiandrosterone EIA (80%) and the Testosterone EIA (232%, Table 1). Conversely, one female and one male showed an increase in the uPM concentrations of more than 90% post GnRH administration in the Progesterone EIA (96%, and 136% respectively), while two females showed an increase in the uPM concentrations of more than 90% in the 5α-Progesterone EIA (218 ± 153%; Table 1). Therefore, due to the limited number of animals showing a response to the administration of GnRH, our results suggest that urine may not be a suitable matrix for the quantification of either androgen or progestogen metabolites. Thus, no further analysis was conducted on the urine samples collected during the saline administration period.

For faecal samples, two of three individuals of both females and males exhibited an increase over 90% in fAM concentrations post GnRH administration for the Epiandrosterone EIA (females: 210 ± 123% and males: 184 ± 30%; Table 2). Conversely, only one individual female and male each showed a response in their fAM concentrations of more than a 90% increase post GnRH administration in the Testosterone EIA (641%, 221%, respectively; Table 2). Thus, the Epiandrosterone EIA was selected as the most suitable assay for measuring fAM concentrations in both male and female naked mole-rats. Also, two of the three females had a response greater than 90% post GnRH injection in the 5α Progesterone EIA (103 ± 7%), but only one female had a response greater than 90% in the Progesterone EIA (153%, Table 2). Thus, the 5α-Progesterone EIA was selected as the best-performing EIA for measuring fPM concentrations in females. Only one male showed an increase in fPM concentrations greater than 90% post injection in the 5α-Progesterone EIA (182%, Table 2).

### 3.2. GnRH and Saline Administration: A Comparison of fAM Concentrations

Overall, for female naked mole-rats, the linear mixed-effects model explained 88% (conditional R^2^) of the variation in fAM concentrations, while the interaction between time and administration explained 2% (marginal R^2^) of the variation in fAM concentrations. In addition, 86% of the variation in fAM concentrations is due to variations between individuals. Finally, the effect of time post injection on fAM concentrations was not dependent on the administration (F = 1.0016, df = 13, and *p* = 0.446).

Similarly, for male naked mole-rats, between-individual variations explained most of the variation in fAM concentrations (89%) as the conditional R^2^ was 91% and the marginal R^2^ (the variation explained by the interaction between time and administration) explained 2% of the variation in fAM concentrations. In addition, the effect of time post injection on fAM concentrations was not dependent on the administration (F = 1.7027, df = 13, and *p* = 0.053).

The results from the linear mixed-effects model for both females and males are likely due to between-individual variations in fAM concentrations (Figure 1, Table 3). There were between-individual variations in baseline and peak fAM concentrations (Figure 1). Whereas in some, no response to the administration of GnRH was detected (Figure 1), and there were between-individual variations in the time at which the peak occurred (Figure 1, Table 3).

### 3.3. GnRH and Saline Administration: A Comparison of fPM Concentrations

The linear mixed-effects model for females explained 86% (conditional R^2^) of the variation in fPM concentrations, and the interaction between time and administration explained 4% (marginal R^2^) of the variation in fPM concentrations. Thus, 82% of the variation in female naked mole-rat concentrations of fPMs is due to between-individual variability. There was a significant effect of time on fPM concentrations which was dependent on the administration (F = 1.9074, df = 13, *p* = 0.025). However, due to the low explanatory power the interaction between time and administration has on fPM concentrations, this significant difference was not considered. As with the fAM concentrations, there are between-individual variations in baseline and peak fPM concentrations. Some animals showed a response to the GnRH administrations, whereas others did not show a response (Figure 2). Furthermore, individuals differed in the time at which the peak occurred (Table 3).

Similarly, the linear mixed-effects model for males explained 83% (conditional R^2^) of the variation in fPM concentrations, and the interaction between time and administration explained 2% (marginal R^2^) of the variation in fPM concentrations. Thus, 81% of the variation in male naked mole-rat concentrations of fPMs is due to between-individual variability. In addition, the effect of time post-injection on fPM concentrations was not dependent on the administration (F = 0.8877, df = 13, and *p* = 0.565). There were between-individual variations in baseline concentrations; however, no males had an increase in fPM concentrations post GnRH administration (Table 3).

## 4. Discussion

In this study, we administered exogenous GnRH to naked mole-rat disperser morphs to validate EIAs that quantify androgen and progestogen metabolites. We tested two EIAs for each hormone class (progestogens: Progesterone and 5α-Progesterone EIA; androgens: Testosterone and Epiandrosterone EIA). The Epiandrosterone EIA [34] and the 5α-Progesterone EIA [32] were selected as the most suitable EIAs for quantifying androgen and progestogen metabolites in the faeces of naked mole-rats, respectively. Interestingly, our results suggest that urine may not be a suitable matrix for measuring either androgen or progestogen metabolites using any of the tested enzyme immunoassays. In addition, there were high between-individual variations in both baseline and peak fAM and fPM concentrations, potentially indicating reduced sensitivity to the administration of exogenous GnRH. We also found a high level of between-individual variation in the time at which the peak occurred after the administration of the GnRH.

To our knowledge, this is the first study to validate EIAs for the measurement of faecal androgen and progestogen concentrations in the naked mole-rat. The use of faeces as a hormone matrix may be a useful tool for studying the endocrinology of the family Bathyergidae in which serial blood sampling is limited because of their small size. In this study, we used physiological validation through the administration of GnRH, which stimulates the hypothalamic–pituitary–gonadal axis, stimulating the secretion of progestogens and androgens by the gonads. For example, in a study investigating the ovarian cycle in Indian rhinoceros (*Rhinoceros unicoris*), GnRH administration led to an increase in luteinizing hormone concentrations in urine [39]. Additionally, this approach was previously successfully used to stimulate the production of androgens in the testes of spinifex hopping mice (*Notomys alexis*) [40] and domestic/house mice (*Mus musculus f. domesticus*) [41] to validate the monitoring of androgens in faeces. Both studies showed significant increases in fAM concentrations following GnRH administration. The results of our study show an increase in fAM concentrations following a GnRH injection for responsive naked mole-rat males and females, indicating that faeces may be a suitable matrix for non-invasive androgen metabolite quantification.

Previous studies used urine as a hormone matrix to non-invasively measure progestogens and androgens in the urine of Damaraland mole-rats (*Fukomys damarensis*), using commercially available kits [42]. In naked mole-rats, glucocorticoids, progestogens, and androgens were quantified in naked mole-rat urine using radioimmunoassays [43,44] that were previously validated by Faulkes et al. [26,45]. However, the assays we evaluated to quantify progestogen and androgen metabolites in urine revealed suboptimal results. The variability in assay suitability is potentially due to the difference in antibody specificities. To see whether hormones are present in naked mole-rat urine, a radiometabolism could be conducted which would provide more clarity regarding the presence and relative abundance of immunoreactive progestogen and androgen metabolites in the urine. However, in both urine and faeces, the variability may be due to the relatively low sample size or unaccounted-for factors such as age and diet. Thus, a replication study with minimum of six individuals for each sex or repeated studies of the same individuals are required to determine if the results can be replicated.

Interestingly, we found high levels of individual variation in the baseline and peak responses in fAM and fPM concentrations after the GnRH injection. This may suggest that some naked mole-rat individuals have a reduced sensitivity of the pituitary to the administration of exogenous GnRH despite being separated from the queen. Faulkes et al. [26,45] administered differing doses of GnRH to naked mole-rats and measured plasma luteinising hormone concentrations and found that the non-breeding females and males were less sensitive to GnRH administration compared to the breeding females and males, respectively. Similar results were found in the eusocial Damaraland mole-rat species whereby Boomsma and Gawne [46] found no significant difference in the plasma LH-concentrations of non-reproductive colony members pre and post administration of exogenous GnRH. However, reduced sensitivity to GnRH is possibly not the only model for reproductive suppression in social African mole-rats. For instance, in cooperatively breeding highveld mole-rats (*Cryptomys hottentotus pretoriae*), there was a significant increase in plasma LH levels for both reproductive and non-reproductive males and females after GnRH administration [47,48]. Likewise, in cooperatively breeding Mashona mole-rats (*F. darlingi*), there was a significant increase in plasma LH concentrations post GnRH administration in both reproductive and non-reproductive males and females [49].

These results suggest that in naked mole-rats, reproductive suppression may be due to a reduced pituitary sensitivity to GnRH administration as well as an inhibition of the release of GnRH from the arcuate nucleus in subordinates. This is supported by the results of Molteno et al. [50], who found that non-reproductive, subordinate Damaraland mole-rat females had higher GnRH concentrations compared to breeders but intense immunoreactivity to GnRH in the median eminence, which may be linked to failure to induce ovulation. However, previous evidence has suggested that subordinate individuals become reproductively viable after separation from the queen [27]. However, our results potentially indicate that for some individuals, reduced GnRH sensitivity persists, even when separated from the queen. One limitation in our study is that there was a variation in the time that the animals were separated from the queen, and information about the ages of the individuals was unavailable. Nevertheless, age and time separated from the queen are unlikely to have significant effects on GnRH sensitivity since the animals have been reported to show insignificant reproductive senescence [51]. Another possible explanation for the reduced GnRH sensitivity in some individuals could be that the GnRH was administered while the animals were housed individually. Blecher et al. [52] found that subordinate females had significant increases in plasma progesterone concentrations when paired with an unrelated individual of the opposite sex compared to when they were separated and still in their natal colony. Future studies should investigate GnRH sensitivity in subordinate naked mole-rats via the administration of GnRH while the animals are in various housing scenarios. These housing scenarios could include direct contact or olfactory contact with unrelated individuals of the opposite sex.

## 5. Conclusions

Through a physiological validation via the administration of GnRH, this study successfully identified suitable EIAs for the non-invasive measurement of androgen and progestogen concentrations in naked mole-rat faeces. Our results suggest that urine may not be a suitable matrix for the measurement of androgen and progestogen metabolite concentrations. Interestingly, the results suggest individual variations in baseline and the response to GnRH administration which are possibly due to some subordinate individuals having a reduced reproductive capacity even when separated from the queen. This study has its limitations in using a relatively low sample size of individuals of varying ages. Thus, further research with a larger number of individuals with consistent age and time separated from the queen is required to fully understand how subordinate naked mole-rats respond to GnRH administration following a significant period of separation from the natal colony.

## Figures and Tables

**Figure 1 animals-13-03039-f001:**
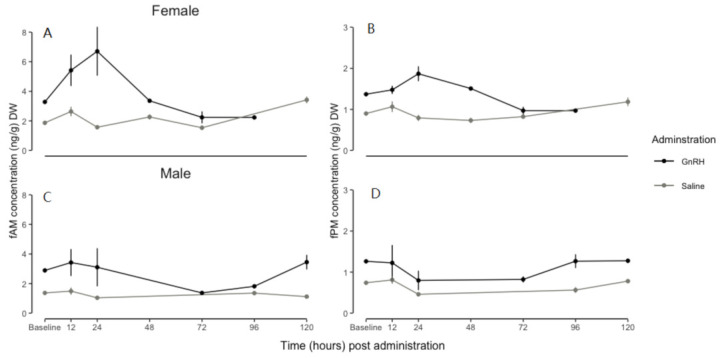
Maximum peak increases in faecal androgen metabolite concentrations (fAM, ng/g DW) for naked mole-rats (*Heterocephalus glaber*) that showed a clear response ((**A**) = female and (**C**) = male; *n* = 1 each) and no response ((**B**) = female and (**D**) = male; *n* = 1 each) to GnRH (2.0 µg of exogenous GnRH in 200 µL of sterile isotonic saline) and saline (200 µL of sterile physiological saline) for each time interval. Points = peak point during time interval.

**Figure 2 animals-13-03039-f002:**
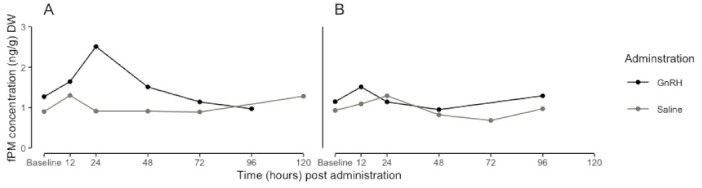
Maximum peak increases in faecal progestogen metabolite concentrations (fPM, ng/g DW) of female naked mole-rats (*Heterocephalus glaber*) that showed a clear response ((**A**), *n* = 1) and no response ((**B**), *n* = 1) to GnRH (2.0 µg exogenous of GnRH in 200 µL of sterile isotonic saline) and saline (200 µL of sterile physiological saline) for each time interval. Points = peak point during time interval.

**Table 1 animals-13-03039-t001:** Individual increases (%) post GnRH administration, reflected in urinary androgen metabolite and urinary progestogen metabolite concentrations for the six female and five male naked mole-rats (*Heterocephalus glaber*) in relation to individual baselines (medians for values from before and 48 h after administration) for each EIA within the first 48 h. Numbers in bold show a percentage change > 80, ^ indicates a peak within 6 h post administration, * indicates a peak within 6–12 h post administration, and ‘ indicates a peak within 12–24 h post administration; no symbol indicates a peak within 24–48 h post administration.

ID	Testosterone EIA	Epiandrosterone EIA	Progesterone EIA	5α Progesterone EIA
F1	* 23	* −24	* 5	* 12
F2	^ −14	^ 28	^ 16	^ −0
F3	* −65	* −67	* 70	* 72
F4	^ 34	^ 70	**^ 96**	**^ 327**
F5	^ 29	^ 55	^ 35	**^ 110**
F6	^ −23	^ 6	^ 23	^ 15
M1	−89	−84	−87	−89
M2	−15	−22	−16	−6
M4	*** 232**	*** 80**	**‘ 136**	‘ 8.5
M5	^ 10	^ −17	^ 75	^ 78
M6	23	35	‘ 1	31

**Table 2 animals-13-03039-t002:** Individual increases (%) post GnRH-administration, reflected in faecal androgen metabolite and faecal progestogen metabolite concentrations for three female and three male naked mole-rats (*Heterocephalus glaber*) compared to individual baseline concentrations (medians for values from before and 48 h after administration) for each EIA within the first 48 h. Numbers in bold show a percentage increase > 90, ^ indicates a peak within 6 h post administration, * indicates a peak within 6–12 h post administration, and ‘ indicates a peak within 24–48 h post administration.

ID	Testosterone EIA	Epiandrosterone EIA	Progesterone EIA	5α Progesterone EIA
F4	* −2	‘ 41	‘ 16	‘ 34
F5	**‘ 641**	**‘ 298**	**‘ 153**	**‘ 109**
F6	^ 69	**^ 123**	^ 67	**^ 98**
M4	*** 221**	**206**	‘ 11	* 41
M5	^ 41	**^ 163**	**^ 182**	*^ 73*
M6	‘ 39	‘ 42	‘ 23	‘ 13

**Table 3 animals-13-03039-t003:** The faecal androgen metabolite (fAM) and faecal progestogen metabolite (fPM) baseline concentrations (ng/g DW), the time post injection until peak (in hours), the peak concentrations (ng/g DW), and % change values for all naked mole-rats (*Heterocephalus glaber*) during the GnRH and saline administrations. Bold indicates percentage increase > 90% and thus responsive individuals. F1–F6 indicate individual females and M1–M6 indicate individual males used for this experiment.

**fAM**	GnRH				Saline			
	Baseline (ng/g DW)	Time	Peak (ng/g DW)	% Change	Baseline (ng/g DW)	Time	Peak (ng/g DW)	% Change

**F1**	**0.45**	**25**	**1.09**	**142**	**1.31**	12	1.91	46
**F2**	**1.33**	**10**	**2.77**	**108**	**1.57**	16	1.68	7
F3	0.95	11	1.43	51	1.11	12	1.16	5
F4	2.25	25	3.17	41	1.85	14	1.44	−22
**F5**	**2.93**	**27**	**11**	**275**	**1.89**	5	3.34	77
**F6**	**0.11**	**4**	**0.24**	**123**	**0.21**	2	0.27	29
M1	2.79	9	4.35	56	3.01	6	5.23	74
M2	3.04	15	4.45	46	3.19	17	5.6	76
**M3**	**1.43**	**14**	**3.7**	**159**	1.25	14	1.36	9
**M4**	**1.82**	**20**	**5.59**	**208**	1.26	10	2.08	65
**M5**	**0.16**	**4**	**0.4**	**150**	**0.26**	**17**	**1.18**	**363**
M6	1.64	19	2.32	42	**1.45**	**27**	**2.98**	**106**
**fPM**	GnRH				Saline			
	Baseline (ng/g DW)	Time	Peak (ng/g DW)	% Change	Baseline (ng/g DW)	Time	Peak (ng/g DW)	% Change

F1	0.86	25	0.93	8	1.18	18	1.43	21
F2	1.81	22	1.56	−14	1.05	16	1.81	72
F3	1.15	1	1.51	32	0.93	20	1.29	39
F4	1.81	28	2.42	34	1.57	14	1.65	5
**F5**	**1.27**	**26**	**2.51**	98	0.9	5	1.3	44
**F6**	**0.26**	**4**	**0.51**	98	0.26	0	0.44	69
M1	1.35	9	1.37	1	1.35	10	1.83	36
M2	1.9	9	1.68	−12	1.83	17	2.42	32
M3	0.74	14	1.01	36	0.68	7	0.46	−32
M4	1.17	10	1.74	48	0.66	10	0.98	48
M5	0.14	4	0.23	70	0.32	21	0.55	75
M6	1.42	13	1.6	13	1.14	6	1.66	46

## Data Availability

The data generated and subsequently analysed during this study will be sent by the corresponding author upon request.

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
