# Peer review of "Non-Invasive Quantification of Faecal and Urine Reproductive Hormone Metabolites in the Naked Mole-Rat (Heterocephalus glaber)"

_animals, 2023, doi:10.3390/ani13193039_

Round 1

Reviewer 1 Report

Majelantle et al., manuscript explores a technique to evaluate the hormone metabolites on feces and urine. The paper is really well written, easy to follow, the methods are very clear. The results are well presented and the discussion is clear.

i have two suggestions, although the authors presents the results of each animal on the table it will be great to see the dispersion of all the samples together, something similar to what you showed on the graphs

Chris Faulkes analyzed hormones back in the day but It will be fantastic to analyze the changes in the hormones after remove animals from the colony and analyze cycles using your method. 

Reviewer 2 Report

GENERAL

This methodological study aimed at establishing EIAs to measure non-invasively steroids in faeces of naked mole-rats. If I would start a study which would need such measurements, I would trust this report and choose the assays tested and recommended here.

MAIN COMMENTS

Introduction: It misses a statement of 1-2 sentences why non-invasive method and not blood is preferred / important.

Methods: Its not clear why the study focussed on dispersers. How were dispersers identified? If I read it correctly, they were housed singly and then changed morphology and showed indication that they would like to escape from their box? I am not sure if (for the current study) it is necessary to label them as “dispersers” and not simply as “single housed” individuals.

L163ff It is not clear to me which kit, which antibody you used: company name etc. This is the most crucial information for such a methodological paper. If somebody wants to use the same method an any mole-rat, what kit / antibody? So its not sufficient to cite other papers here to get and then search for the information, but you must be more detailed.

L170 how many replicates? The controls were from the kit? Or did you create your own controls from NMR samples, which would be the more appropriate way to estimate the assay variation of your method.

L176 As this is a methods paper, showing these serial dilution curves would be appreciated.

L216 and elsewhere: You must in the introduction or methods section clearly explain what increase in hormone levels in % you expect. Is it normally >>100%? Why is the 100% / 8ß% threshold you mention relevant?

L216ff This part can be shortened a lot and with this make much clearer, by simply stating mean + SD and test statistics in one sentence per hormone and referring to table 1. Or include mean and test statistics in Table 1 and simply state in the text “Urine hormone concentrations were not affected significantly by the treatment (Table 1) and therefore no further analysis was conducted on the urine samples.” Results for faeces can then be similarly compressed.

L311-313 This is correct, but another study with a larger sample size might find no difference between the assays. Your selection was – due to small sample size – mainly based on descriptive statistics, which should be stated.

In general, I have seen huge variation in steroid hormone metabolites in faecal samples in different species, which remains largely unexplained and ignored by the literature. Its possible if you would replicate the study with exactly the same individuals, you might get different results, or the same results, but this time different individuals showing a clear increase in metabolites. Thus, shortly discussing a study with (i) larger sample size and (ii) repeatedly using the same individuals to test whether “they” (i.e. their faecal samples) always show the same response would be good.

Similarly L350: Without doing the experiment at least twice with each individual, you cannot be 100% sure its individual variation you observed, i.e. whether its consistent within individuals (as in personality) or due to some unknown factors influencing metabolite concentration in the faeces, including but not restricted to: how / when you collected the sample, unrecognised contamination with urine, ambient temperature while sample was collected, processing time, differences in microbiome, difference in passage time due to the time and amount of food the animal was eating.

I know its irritating, but from the data I have seen, metabolic hormone concentrations in faeces care extremely variable, for unknown reasons.

L371 overall, your study would suggest that subordinates are not supressed by the breeders, but they themselves refrain from investing in activation of the gonadal axis as long as there is no opportunity to breed = they met a potential mate. In fact, dispersal is very expensive, especially in NMR, and success of dispersal determines reproductive success, so all energy should be invested into this. And only after successful dispersal = having found a mate, does investment into activation of the gonadal axis make sense.

MINOR COMMENTS

See also edits in attached PDF

I don’t like to read abbreviations such as NMR instead of naked mole-rat throughout the manuscript (though I did the same with the manuscripts from my PhD and still do it internally for my current study species, but not in publications).

Google tells me” NMR is an abbreviation for Nuclear Magnetic Resonance.”

L12 /L 26 of both sexes?

L18 /L32 Why? What were the results?

L23 / L40 for how long?

L36 less than 100 can be anything between 0 (or even less) to 99. So please + SD

L48 and elsewhere “reproductive capacity” is a term I am not used to, and when I google it it is mainly used as a measure how many eggs a female (insect) can produce. I would try to avoid it or at least define it.

First paragraph introduction: When referring to previous studies, always use the same tense (present or past). I changed it to present now (as that was the first tense you used) but you might prefer past tense, and you should be consistent throughout the manuscript.

L62 sentence not clear

L70 There is a sentence missing: What happens between secretion into the blood until metabolites are in urine / faeces

L77 add reference

L78 add Clutton-Brocks science paper as reference

Clutton-Brock, T. 2021. Social evolution in mammals. Science, 373, eabc9699.

L80 adult individuals?

L139 – L146 move to section 2.5

L184 Why the threshold of 0.05 mg/ml?

L190: It is not clear to me what you are saying here. Did you test different EIA kits? (See main comment on more information on the kits / antibodies used).

L199-201 rewrite this complex sentence so one can understand it

L205 did you use total concentration or the relative ones obtained from equation (1)?

L266 consider rephrasing, that you found no response, not that the animal did not show one (maybe it did, but you did not get the correct sample at the correct time, or there was a problem during the assay).

L317 you mean “indicating individual variation in responsiveness”? Are there alternative explanations, like body mass, how much each individual did eat?

L335-337 I don’t understand, is this necessary?

L346 Maybe delete this last sentence. Is there any doubt that steroids are present in urine? To test for the potential antibody problem, first analysing plasma would help. You had very few samples. In how many assays were they analysed? Is it possible that one assay did not work, and that’s the reason for the results? Or did you try repeatedly?

L353 Did Faulkes measure in serum?

L370 immunoreactivity where?
